# ImmCellTyper facilitates systematic mass cytometry data analysis for deep immune profiling

Jing Sun[1]*[†], Desmond Choy[2†], Nicolas Sompairac[2], Shirin Jamshidi[2], Michele Mishto[1,3], Shahram Kordasti[2,4,5]*

[1]Centre for Inflammation Biology and Cancer Immunology & Peter Gorer Department of Immunobiology, King's College London, London, United Kingdom; [2]School of Cancer and Pharmaceutical Sciences, King's College London, London, United Kingdom; [3]Research Group of Molecular Immunology, Francis Crick Institute, London, United Kingdom; [4]Haematology Department, Guy's Hospital, London, United Kingdom; [5]Department of Clinical and Molecular Sciences, Università Politecnica delle Marche, Ancona, Italy

*For correspondence:
jing.1.sun@kcl.ac.uk (JS);
shahram.kordasti@kcl.ac.uk (SK)

[†]These authors contributed equally to this work

Competing interest: The authors declare that no competing interests exist.

**Abstract** Mass cytometry is a cutting-edge high-dimensional technology for profiling marker expression at the single-cell level, advancing clinical research in immune monitoring. Nevertheless, the vast data generated by cytometry by time-of-flight (CyTOF) poses a significant analytical challenge. To address this, we describe ImmCellTyper (https://github.com/JingAnyaSun/ImmCellTyper), a novel toolkit for CyTOF data analysis. This framework incorporates BinaryClust, an in-house developed semi-supervised clustering tool that automatically identifies main cell types. BinaryClust outperforms existing clustering tools in accuracy and speed, as shown in benchmarks with two datasets of approximately 4 million cells, matching the precision of manual gating by human experts. Furthermore, ImmCellTyper offers various visualisation and analytical tools, spanning from quality control to differential analysis, tailored to users' specific needs for a comprehensive CyTOF data analysis solution. The workflow includes five key steps: (1) batch effect evaluation and correction, (2) data quality control and pre-processing, (3) main cell lineage characterisation and quantification, (4) in-depth investigation of specific cell types; and (5) differential analysis of cell abundance and functional marker expression across study groups. Overall, ImmCellTyper combines expert biological knowledge in a semi-supervised approach to accurately deconvolute well-defined main cell lineages, while maintaining the potential of unsupervised methods to discover novel cell subsets, thus facilitating high-dimensional immune profiling.

## eLife assessment

ImmCellTyper presents a **useful** toolkit for CyTOF data analysis, integrating BinaryClust for semi-supervised clustering and cell type annotation. The evidence supporting the findings is **convincing**, with appropriate and validated methodology. This tool will be helpful to researchers in immunology and cytometry, offering a robust solution for cell type identification and differential analysis.

## Introduction

Mass cytometry or cytometry by time-of-flight (CyTOF) is a powerful high-throughput single-cell technology which employs stable elemental isotopes, as the same manner of fluorophores, to detect cellular proteins of interest. This approach successfully tackles the panel multiplex challenges faced

by traditional flow cytometry due to spectral overlap and permits simultaneous measurement of over 40 parameters on millions of cells. To date, CyTOF has been widely applied in basic and translational medical research, such as deep immunophenotyping and characterisation of novel refined cell subsets, immune monitoring of cell-adoptive therapy, and dissecting cell subpopulations from heterogeneous tumour samples (*Spitzer and Nolan, 2016*). Nonetheless, the advantages of CyTOF come with the problem of handling high-dimensional dataset. Traditional gating strategy for flow cytometry, while still serving as the 'gold standard' for cell population identification in cytometry data, may not be an optimal option for CyTOF data due to its high-dimensional settings. The high-parametric resolution in CyTOF, aimed at revealing previously undiscovered cell subpopulations, leads to a significant increase in the complexity of gating schemas and hierarchical depth, which makes manual gating extremely labour intensive and time-consuming (*Kimball et al., 2018*). Therefore, effective computational tool-kits and pipelines are entailed for CyTOF data mining and analysis.

Efforts have been directed towards developing means of clustering algorithms to deconvolute a pool of live cell mixture into distinct cell populations, which facilitates CyTOF data analysis. For instance, unsupervised methods, which include flowSOM (*Van Gassen et al., 2015*), Phenograph (*Levine et al., 2015*), X-shift (*Samusik et al., 2016*), spade (*Qiu et al., 2011*), DensVM (*Becher et al., 2014*), etc., often combine with dimension reduction techniques like *t*-distributed stochastic neighbour embedding (*t*-SNE) (*Maaten and Hinton, 2008*), Uniform Manifold Approximation and Projection (UMAP) (*McInnes and Healy, 2018*), principal component analysis (PCA), etc., and require manual annotation of each cluster based on the marker expression patterns indicated by heatmaps. This approach works well to analyse populations in a data-driven manner, with all files concatenated and analysed all at once. Compared to manual gating, it has advantages in terms of convenience, efficiency, and relative unbiasedness from biological preconception, facilitating the detection of novel cell phenotypes for deep phenotyping. However, the unsupervised approaches are not always ideal and suitable for cytometry data. Mathematical clustering does not necessarily have biological meaning, leading to occasional inaccuracies. And several benchmarking studies suggested that the accuracy for these unsupervised tools may not be optimal (*Liu et al., 2019*). Additionally, the technical uncertainty of results produced by different clustering approaches remains a conundrum. Even for the same unsupervised method, the discrepancy among different runs without setting a seed, reduces the reproducibility and may cause confusion, particularly for biologists with limited computational knowledge. Moreover, manual validation is also essential, as biological annotation is the inevitable step to provide biological relevant labels for the clusters. However, this process can be time-consuming and subjective, hindering the automation of pipelines. This problem is particularly pronounced for a large marker panel and samples with high heterogeneity, resulting in a higher number of clusters that need to be annotated.

Advances in artificial intelligence have accelerated the development of alternative clustering methods in a supervised manner for cell type inference. These methods consider the 'ground truth' or prior knowledge about the marker expression of each given cell types to automatically label each cell. Currently, a couple of semi-automatic methods have been developed, such as linear discriminant analysis (LDA) (*Abdelaal et al., 2019*), DGCyTOF (*Cheng et al., 2022*), CyAnno (*Kaushik et al., 2021*), DeepCyTOF (*Li et al., 2017*), Automated Cell-type Discovery and Classification (ACDC) (*Lee et al., 2017*), Semi-supervised Category Identification and Assignment (SCINA) (*Zhang et al., 2019*), etc. ACDC (*Lee et al., 2017*) and SCINA (*Zhang et al., 2019*) use a matrix of pre-defined markers for each cell type to annotate the clusters showing the same signature. These methods assume that markers are either expressed or not expressed (binary), which limits their ability to distinguish cell subtypes with similar phenotypes, particularly non-canonical cell types that cannot be easily separated linearly. Alternatively, DeepCyTOF and LDA use a marker expression matrix extracted from manually gated cell types as a training dataset to build a machine-learning model for cell type prediction. This approach has higher precision and accuracy compared with aforementioned methods (*Liu et al., 2019*). Nonetheless, it can be labour intensive for preparation of the training set manually. Also, these methods are limited in their ability to predict novel cell subsets beyond the pre-gated set of cell types and lack a systematic and comprehensive way to assign the cells which were not identifiable under any of the gated cell types. New solutions have emerged with algorithms such as DGCyTOF (*Cheng et al.,*

*2022*) and CyAnno (*Kaushik et al., 2021*), the former adopts a deep learning classification combined with hierarchical stable-clustering methods and an iteration calibration system to identify known cell types and assign novel subsets; while CyAnno is based on a machine-learning framework which allows the integrative modelling of both 'gated' and 'ungated' cells. Both methods have demonstrated high accuracy in their test datasets, but unfortunately are not widely used by the research community, possibly due to the issue for the hassle of training data preparation, the lack of user-friendliness and implementation challenges for bench researchers.

To address the common drawbacks of current semi-supervised and unsupervised clustering algorithms and preserve their strengths in discovering both canonical and non-canonical cell subsets, respectively, we propose a strategy implemented in ImmCellTyper for cell classification named BinaryClust. By considering biologists' prior biological knowledge and interpretation for canonical cell clusters in a semi-supervised manner, BinaryClust first automatically characterises the main cell lineages in a fast and accurate way. Subsequently, it extracts specific cell types of interest for further clustering using unsupervised algorithms to identify cell subsets including previously unreported non-conventional population. In addition, this R-implemented pipeline takes advantage of SingleCell-Experiment class for data management, providing an easy-to-use and organised systematic workflow of CyTOF data handling. The whole pipeline includes quality control and batch effect correction, which helps to effectively pool datasets from different batches, and ensures the robustness for downstream analysis. Meanwhile, modules like dimension reduction, semi-supervised and unsupervised clustering (flowSOM and Phenograph), interactive data visualisation, and statistical testing for complex study design were also incorporated in this pipeline. Compared with existing integrated computational workflow, such as CATALYST (*Nowicka et al., 2017*), CapX (*Marsh-Wakefield et al., 2019*), Cytofkit (*Chen et al., 2016*), and ImmunoCluster (*Opzoomer et al., 2021*), which was developed and maintained by our team, this recently developed toolkit advanced further on coherence, functionality and user-friendliness. Overall, this approach has the potential to facilitate and smooth the investigation of CyTOF-based research.

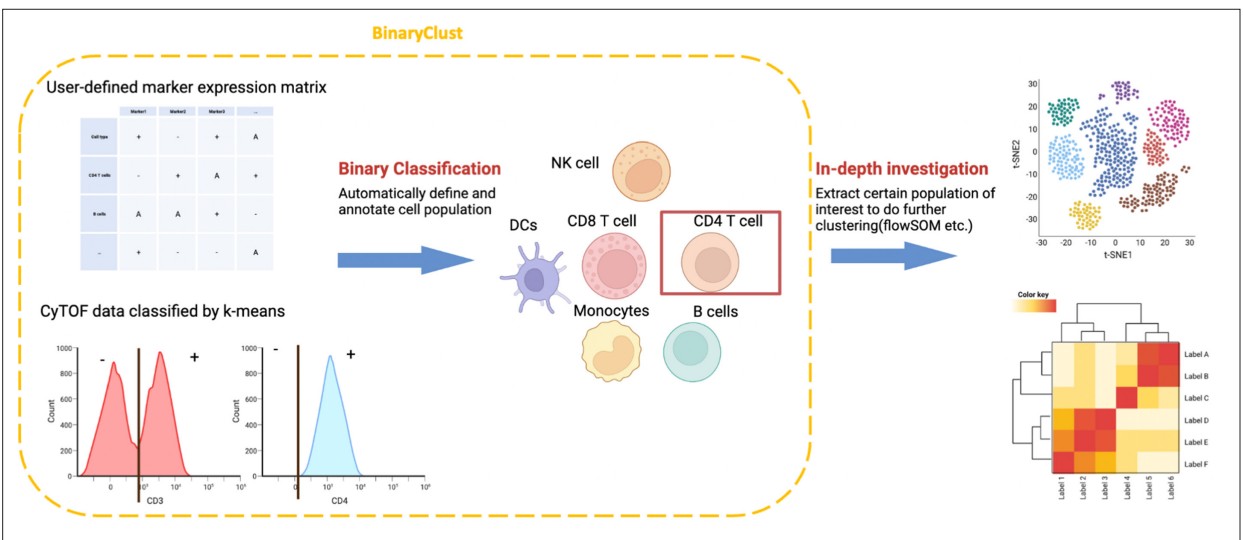

**Figure 1.** Schematic diagram of BinaryClust framework. Semi-supervised classification is first performed on selected markers in the user-defined marker expression matrix to classify and annotate major cell types. Population-of-interest can be further extracted and explored using unsupervised clustering methods followed by differential analysis. Created with BioRender.com.

The online version of this article includes the following figure supplement(s) for figure 1:

**Figure supplement 1.** Manual hierarchical gating strategy for main cell linages from human peripheral blood mononuclear cell (PBMC) samples (myeloproliferative neoplasm [MPN] dataset, *n* = 9).

**Figure supplement 2.** Clean-up procedure of cytometry by time-of-flight (CyTOF) data using Cytobank.

# Results

## BinaryClust is comparable with manual gating in quantifying the abundances of main cell lineages

The core concept for BinaryClust is depicted in *Figure 1*, where a simple user-designed cell type marker expression matrix is required and serves as a reference to characterise the main cell lineages, with positive markers indicated in '+', negative markers in '-', and irrelevant markers in 'A'. *K*-means (*k* = 2) will be applied to divide the positive and negative cell population of each marker, then align it to the reference table to infer main cell types. This is followed by the extraction of population-of-interest for downstream clustering using unsupervised methods for subpopulation discovery.

To assess the performance of the automated cell type classification and prediction, we generated a test CyTOF dataset using peripheral blood mononuclear cell (PBMC) samples from seven patients with myeloproliferative neoplasm (MPN) and two healthy donors, employing a 37-marker deep immunophenotyping panel (*Appendix 1—table 1*). Manual gating was performed by two independent experts which identified seven main cell lineages. These results act as the reference for evaluating the computer-aided methods. The hierarchical sequential gating strategy was explicitly illustrated in *Figure 1—figure supplement 1*. We evaluated the agreement between manual gating results and BinaryClust results regarding cell frequencies of each population. The mean value was calculated from manual gating results of two experts to compare with BinaryClust-generated results using Pearson correlation analysis. As shown in *Figure 2A*, the two methods exhibit a strong correlation with coefficient ($R^2$) equals to 1, 1, 0.99, 0.96, 0.96, 0.99, 0.99 for CD4 T cells, CD8 T cells, dendritic cells, NK cells, monocytes, and gamma delta T cells, respectively (all $p < 0.0001$). And most of the data points remain close to the line of equality (red line, $R^2 = 1$), indicating a high degree of agreement. Meanwhile, the Bland–Altman plots in *Figure 2B* also suggest no consistent bias of manual gating versus BinaryClust across all the identified cell types. The good performance of BinaryClust was further validated in the influenza dataset published by our group (*Alimam et al., 2021*), which contains FCS files from 11 individuals with six main immune cell types detected (*Figure 2—figure supplement 1A, B*).

## BinaryClust achieves high accuracy and speed compared with flowSOM and LDA

To further evaluate BinaryClust's performance, we compared it with the well-performing unsupervised algorithm flowSOM and supervised classifier LDA. FlowSOM was run on the same MPN dataset with *k* value set as 20, followed by manual annotation and cluster merging to identify the same cell populations as in manual gating and BinaryClust. The cell frequencies derived from BinaryClust, flowSOM, and manual gating (expert1 and expert2) were compared using interaction plot (*Figure 3*). We observed that BinaryClust remains consistent with manual gating (all $p > 0.05$), whereas flowSOM identified significantly less gamma delta T cells and dendritic cells ($p < 0.001$ and $p = 0.006$, respectively) compared with the other three measurements. We also increased the initial *k* value to 40 to improve accuracy by over clustering then merging clusters (*Van Gassen et al., 2015*). We found that the average frequency for gamma delta T cells and dendritic cells increased to 1.5% and 4.65%, respectively, but still remained significantly different ($p < 0.05$) from the other two methods. It is also interesting to see that although the same gating strategy was applied, different experts obtained results with slight variation despite no statistical significance. The same findings were confirmed using boxplot as shown in *Figure 3—figure supplement 1*.

Subsequently, LDA was also tested on the same dataset. Since this method requires a training dataset from manual gating to build a model, we exported the cell events and cluster assignment from Cytobank, then equally partitioned all cells into training and test dataset. Due to the difference in method implementation, we did not include LDA for cell abundance quantification comparison with other methods, since only half of the cells were used for prediction. To ensure an equitable comparison and further evaluate the accuracy, *F*-measure, and Adjusted Rand Index (ARI) were calculated using manual gating cluster IDs as reference. As indicated in *Table 1*, for the MPN dataset, BinaryClust achieved high *F*-measure and ARI in all seven cell types with an average of 0.94 and 0.91, respectively. The performance remains excellent in influenza dataset as well (average *F*-measure = 0.98, *Figure 2—figure supplement 1C*). LDA has equivalent prediction accuracy as BinaryClust with an average *F*-measure of 0.93 and ARI of 0.90. Both supervised methods outperformed flowSOM

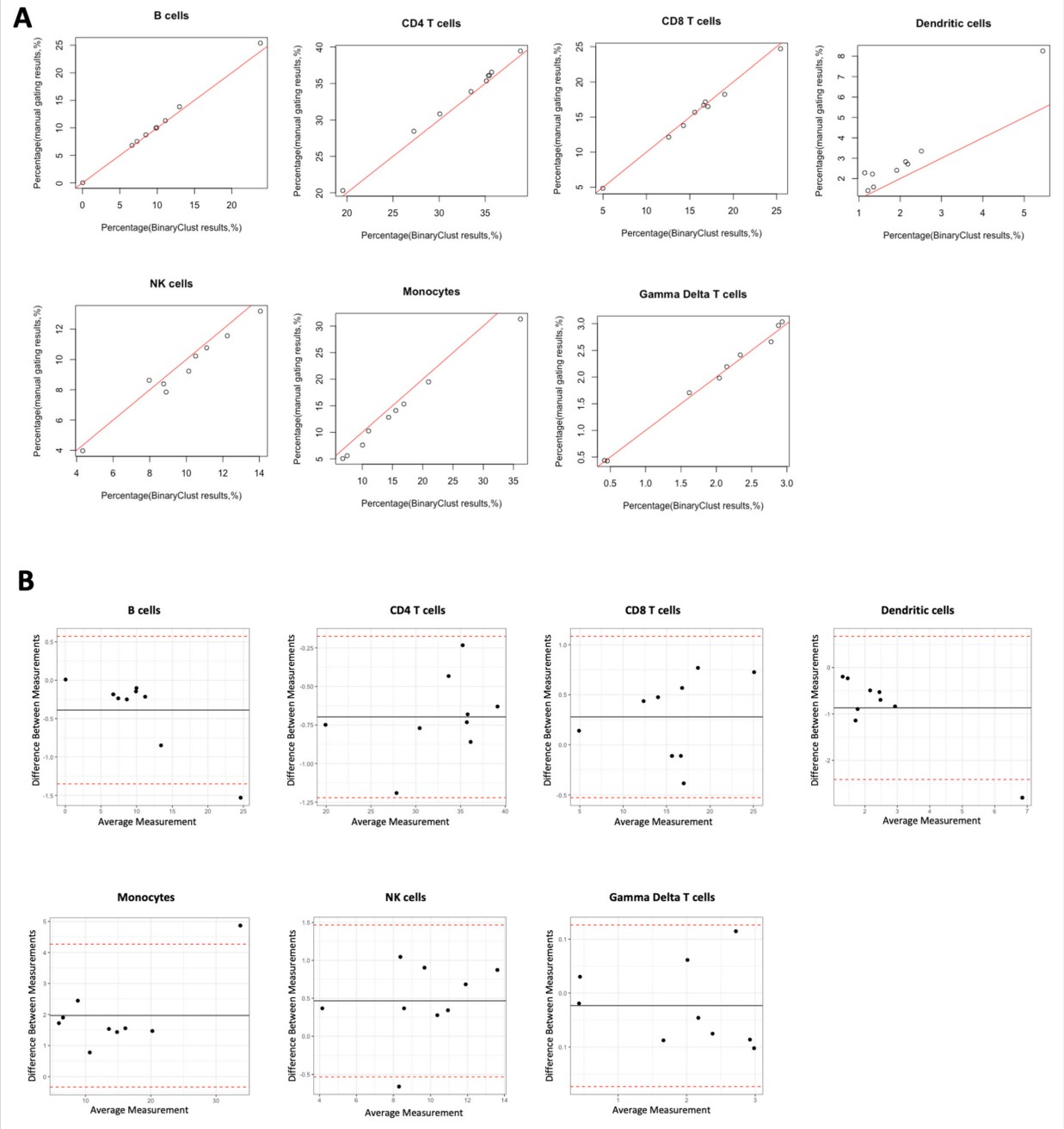

**Figure 2.** Agreement evaluation comparing manual gating and BinaryClust in myeloproliferative neoplasm (MPN) cohort (*n* = 9). Manual gating of B cells, CD4 T cells, CD8 T cells, dendritic cells, NK cells, monocytes, and gamma delta T cells were performed by two independent experts using Cytobank, and mean values of the population percentages were calculated to compare with BinaryClust results. Each dot represents one patient sample. (**A**) Scatter plot showing the correlation between the two methods, with the red line indicating perfect agreement (correlation coefficient = 1). (**B**) Bland–Altman plots of the two measurement methods among all the cell populations, with the black line suggesting the mean observed difference and red dotted lines indicating limits of agreement (1.96× standard deviations).

The online version of this article includes the following figure supplement(s) for figure 2:

**Figure supplement 1.** Agreement evaluation between ImmCellTyper and manual gating in influenza dataset (*n* = 11).

(average *F*-measure = 0.75, average ARI = 0.66), which is in line with previous benchmarking study on evaluating supervised and unsupervised clustering algorithms (*Liu et al., 2019*). Notably, for cell types that constitute a substantial proportion in the pool like CD4 T cells, CD8 T cells, NK cells, monocytes, and B cells, flowSOM can identify them with high precision and sensitivity, while this is not the

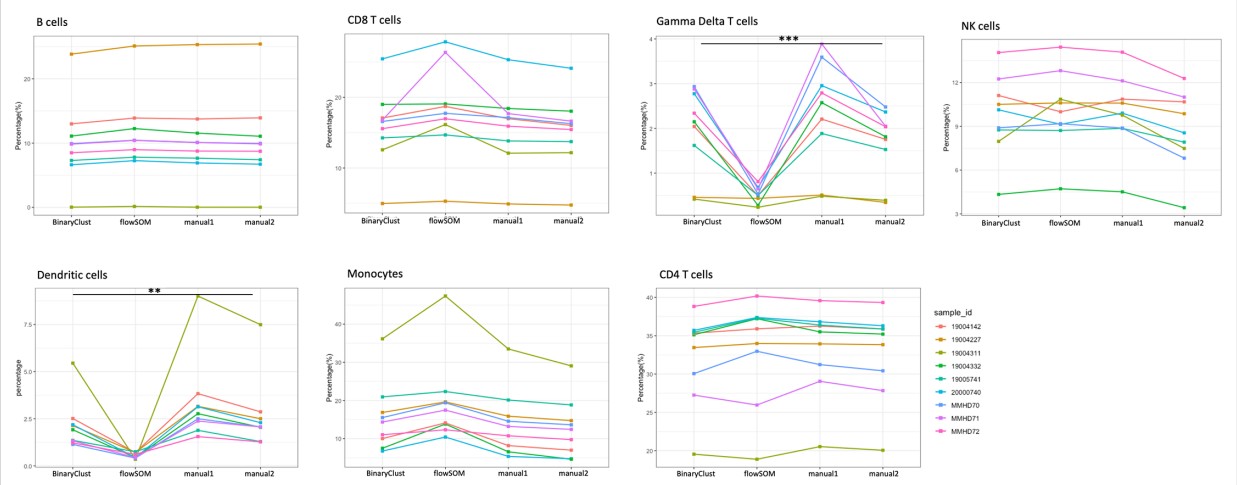

**Figure 3.** Comparison of manual gating (manual1 and manual2), BinaryClust, and flowSOM clustering results in myeloproliferative neoplasm (MPN) cohort (*n* = 9). Interaction plots showing the individual measurement (percentage) of each study participant with indicated colours by different methods across main cell lineages (B cells, CD8 T cells, gamma delta T cells, NK cells, dendritic cells, monocytes, and CD4 T cells); analysis of variance (ANOVA) was used for statistical testing, and significance was marked by asterisk. *p < 0.05, **p < 0.01, ***p < 0.001, ****p < 0.0001.

The online version of this article includes the following figure supplement(s) for figure 3:

**Figure supplement 1.** Boxplots of the indicated cell percentages generated by different methods.

case for dendritic cells and gamma delta T cells which account for less than 3% of the whole population (*F*-measure = 0.51 and 0.15, respectively). Even so, the overall *F*-measure and ARI for flowSOM considering cell proportion reached 0.81 and 0.80.

It has been demonstrated that flowSOM and LDA are among the fastest clustering algorithms without compromising their performance. Here, we compared the speed among the three approaches in MPN, Influenza, and COVID-19 datasets (*Chevrier et al., 2021*) which contain 2,231,053, 210,933, and 3,862,628 cells, respectively. Since LDA requires training data to be executed, we did not run it on COVID-19 dataset due to the absence of manual gating results. As shown in *Figure 4*, BinaryClust exhibited the highest speed in both MPN and Influenza dataset but fell behind flowSOM slightly in COVID-19 dataset.

## ImmCellTyper pipeline supports interactive data visualisation and comparison among study groups

BinaryClust is an important component of ImmCellTyper, a comprehensive integrated pipeline designed for systematic CyTOF data mining. Hence, we utilised the visualisation functions to further prove the reliability and robustness of BinaryClust.

BinaryClust inherently considers CyTOF makers as binary distributed which is in most of the case, but not always. Therefore, it is crucial to check the marker behaviour before running the pipeline. As shown in *Figure 5A*, all markers selected for the classification matrix (*Figure 5B*) displayed a binary distribution implying the suitability for this pipeline. After clustering (*Figure 5C*), median marker expression heatmap (*Figure 5D*) can check the reliability of the results before proceeding for downstream analysis. In MPN dataset, we projected the cluster assignment resulted from BinaryClust and manual gating to UMAP (*Figure 5E*), along with the expression of the phenotypic markers (*Figure 5F* and *Figure 5—figure supplement 1A*). High similarity was observed between BinaryClust and manual gating, whereas slight difference was found on flowSOM results coloured UMAP on islands of CD8 T cells and CD4 T cells (*Figure 5—figure supplement 1B*). FlowSOM appeared to classify cells that were close on spatial distance into the same cluster, in contrast to prior knowledge-based methods: BinaryClust and manual gating.

**Table 1.** Precision, recall, *F*-measure, and ARI of indicated clustering methods.

| | Gated population | Counts | Cluster cell counts | True positive | Precision | Recall | *F*-measure | Average *F*-measure | ARI | Average ARI |
|---|---|---|---|---|---|---|---|---|---|---|
| BinaryClust2 | CD4 T cells | 1,270,041 | 1,232,832 | 1,226,712 | 1.00 | 0.97 | 0.98 | 0.93 | 0.95 | 0.91 |
| | CD8 T cells | 601,995 | 601,639 | 582,402 | 0.97 | 0.97 | 0.97 | | 0.95 | |
| | NK cells | 389,261 | 380,999 | 376,561 | 0.99 | 0.97 | 0.98 | | 0.97 | |
| | Monocytes | 576,624 | 625,735 | 568,924 | 0.91 | 0.99 | 0.95 | | 0.92 | |
| | B cells | 397,279 | 380,170 | 374,501 | 0.99 | 0.94 | 0.96 | | 0.95 | |
| | Dendritic cells | 135,894 | 86,012 | 84,130 | 0.98 | 0.62 | 0.76 | | 0.74 | |
| | TCRgd T cells | 87,194 | 73,599 | 73,562 | 1.00 | 0.84 | 0.91 | | 0.91 | |
| flowSOM | CD4 T cells | 1,270,041 | 1,268,929 | 1,226,712 | 0.97 | 0.97 | 0.97 | 0.70 | 0.90 | 0.66 |
| (*k* = 20) | CD8 T cells | 601,995 | 692,374 | 571,344 | 0.83 | 0.95 | 0.88 | | 0.81 | |
| | NK cells | 389,261 | 393,646 | 365,403 | 0.93 | 0.94 | 0.93 | | 0.91 | |
| | Monocytes | 576,624 | 792,447 | 568,284 | 0.72 | 0.99 | 0.83 | | 0.73 | |
| | B cells | 397,279 | 405,886 | 390,077 | 0.96 | 0.98 | 0.97 | | 0.96 | |
| | Dendritic cells | 135,894 | 20,832 | 18,254 | 0.88 | 0.13 | 0.23 | | 0.22 | |
| | TCRgd T cells | 87,194 | 19,720 | 4967 | 0.25 | 0.06 | 0.09 | | 0.08 | |
| LDA | CD4 T cells | 634,733 | 665,481 | 632,700 | 0.95 | 1.00 | 0.97 | 0.93 | 0.93 | 0.90 |
| | CD8 T cells | 301,138 | 327,469 | 299,309 | 0.91 | 0.99 | 0.95 | | 0.92 | |
| | NK cells | 194,470 | 202,110 | 187,259 | 0.93 | 0.96 | 0.94 | | 0.92 | |
| | Monocytes | 288,338 | 301,387 | 282,836 | 0.94 | 0.98 | 0.96 | | 0.94 | |
| | B cells | 198,582 | 211,090 | 195,908 | 0.93 | 0.99 | 0.96 | | 0.94 | |
| | Dendritic cells | 68,152 | 90,456 | 61,785 | 0.68 | 0.91 | 0.78 | | 0.75 | |
| | TCRgd T cells | 43,405 | 43,064 | 39,976 | 0.93 | 0.92 | 0.92 | | 0.92 | |

ARI, Adjusted Rand Index; LDA, linear discriminant analysis.

## Application of ImmCellTyper pipeline to the COVID-19 dataset demonstrates its versatile functionalities for comprehensive data analysis

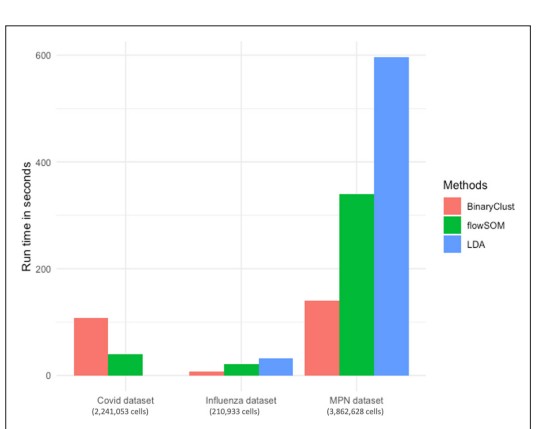

**Figure 4.** Comparison of BinaryClust, flowSOM, and linear discriminant analysis (LDA) on speed. Bar chart showing runtime (in seconds) of the three methods in three different datasets.

To showcase the analytical and visualisation functions of ImmCellTyper pipeline, we used the dataset published by *Chevrier et al., 2021*, which described the immune signature of mild and severe COVID-19 patients in comparison with healthy individuals. There are a total of 82 FCS files with a 40-plex marker panel focussing on innate immunity in this dataset. An initial marker expression check was carried out based on the user-defined matrix (*Figure 6B*) and displayed in *Figure 6A*. BinaryClust was then performed and identified 12-cell populations as expected. We used *t*-SNE in this dataset for dimension reduction, coloured by cell types and faceted by disease conditions, which exhibited substantial

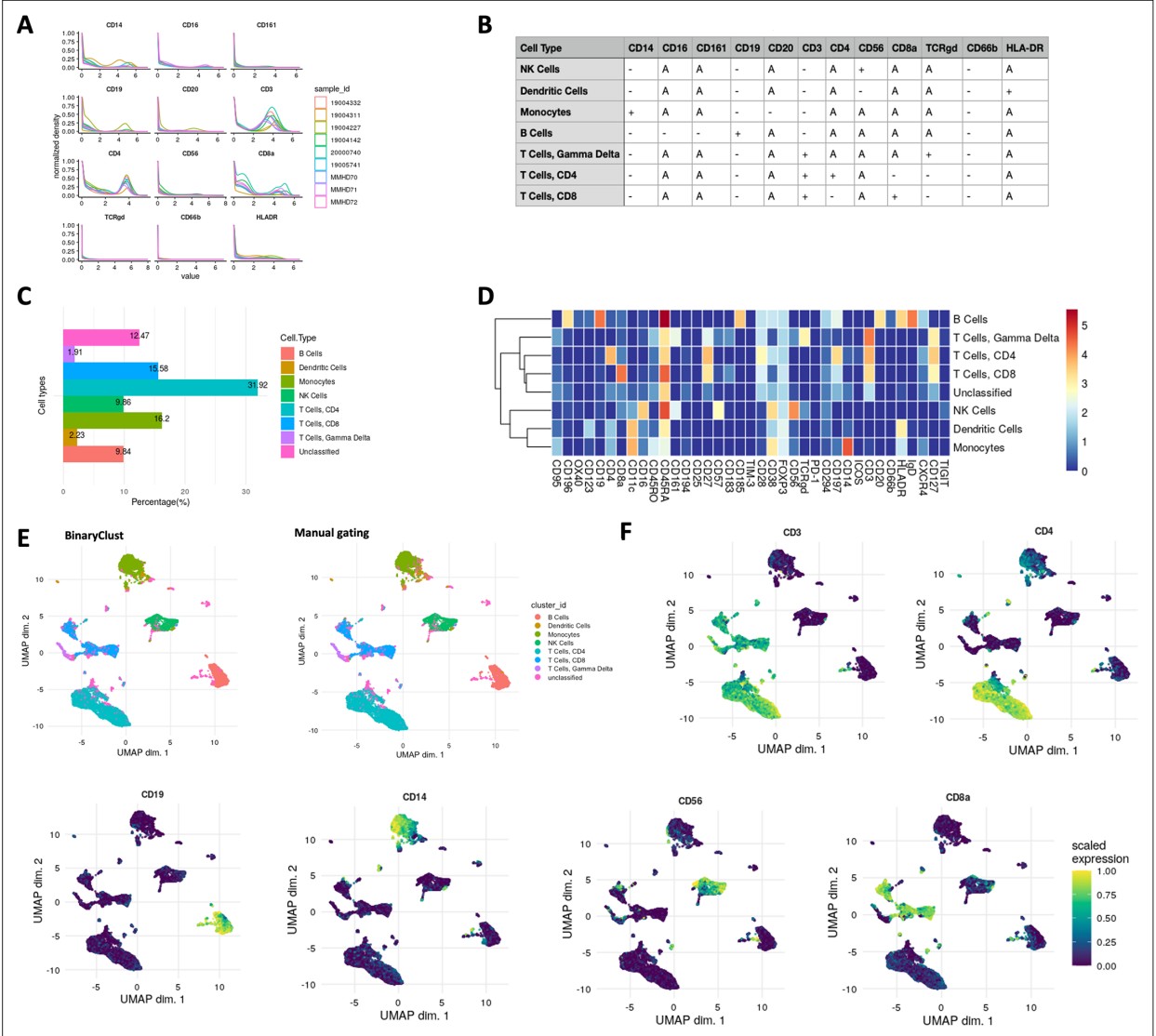

**Figure 5.** Cell type characterisation and visualisation using ImmCellTyper pipeline in myeloproliferative neoplasm (MPN) dataset (*n* = 9). (**A**) Intensity distribution of selected phenotypic markers used for BinaryClust classification, coloured by sample_id. (**B**) Pre-defined expression classification matrix for the MPN dataset, '+' indicates positive, '-' indicates negative, and 'A' suggests 'any'. (**C**) Proportion of the main cell lineages of all cells in the concatenated FCS files after classification. (**D**) Median marker expression heatmap of BinaryClust classification results. (**E**) UMAP plot of random downsample of 2000 cells per patient coloured by main cell types based on BinaryClust classification (left) and manual gating results (right). (**F**) UMAP plots coloured by normalised expression of indicated markers (CD3, CD4, CD8a, CD20, CD19, CD14, and CD56) across 2000 cells per sample.

The online version of this article includes the following figure supplement(s) for figure 5:

**Figure supplement 1.** Comparison of different clustering methods in MPN cohort.

immune alteration among healthy control, mild and severe COVID-19 patients (*Figure 6C*). From the heatmap in *Figure 6D*, we can have an overview of the marker expression of each population, which remains consistent with our initial definition indicated in the expression matrix.

The abundances of the identified cell types were quantified for each individual study participant in the format of stacked histogram (*Figure 7A*) and summarised in boxplot (*Figure 7B*). Compared with healthy volunteers, there are significant immune alterations in COVID-19 patients in B cells, basophils, cDCs, monocytes, NK cells, CD8 T cells, neutrophils, and pDCs (all p < 0.05), which depicts a similar trend to the original paper, despite slight discrepancies caused by different statistical methods used. Kruskal–Wallis test was conducted for this dataset followed by multiple testing correction (Benjamini–Hochberg [BH] procedure) and Dunn's test for post hoc analysis, while Chevirer et al.

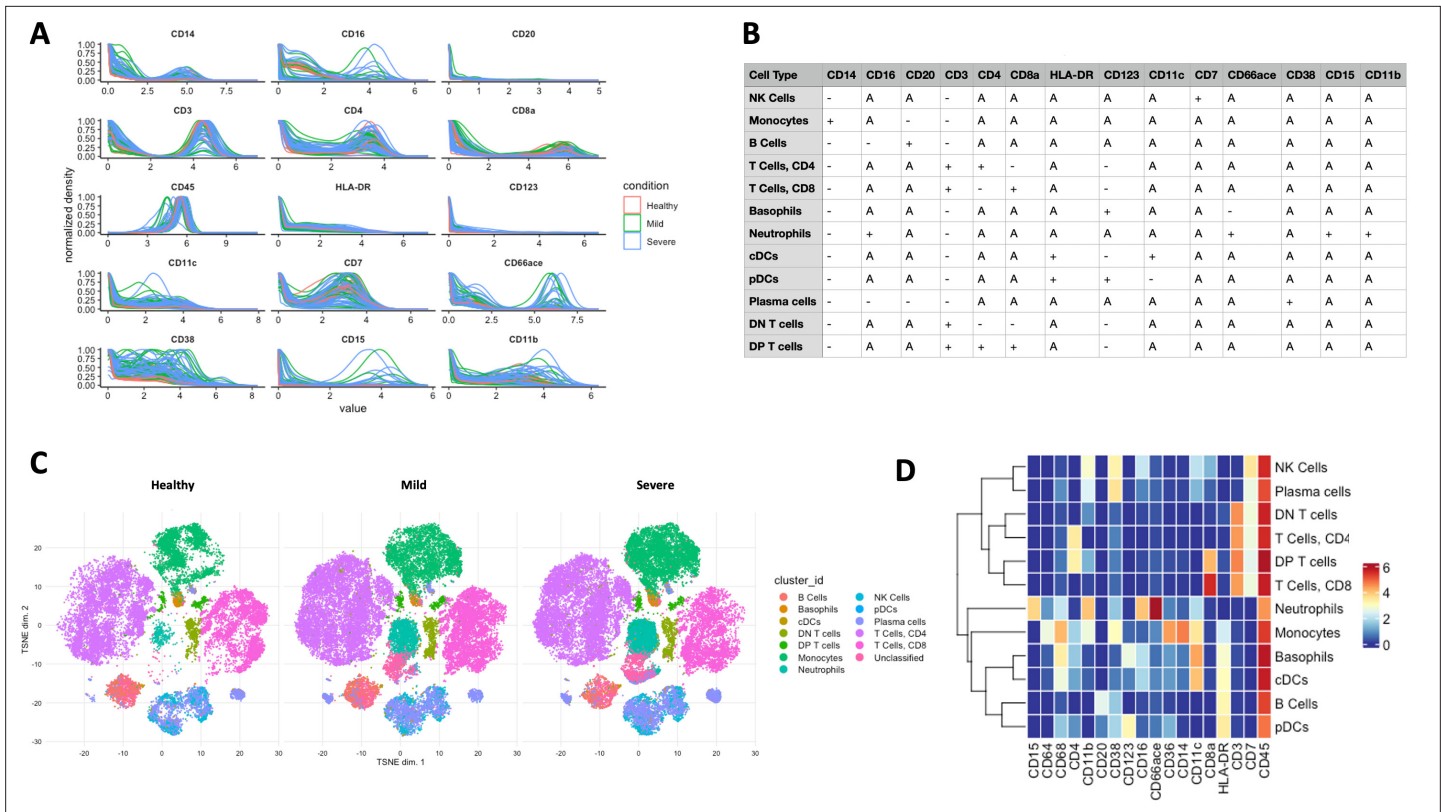

**Figure 6.** Applying ImmCellTyper pipeline on COVID-19 patient dataset (*n* = 82) published by ***Chevrier et al., 2021***. (**A**) Marker intensity distribution of selected phenotypic markers used for BinaryClust classification, coloured by disease severity (*n* = 22 healthy individuals, 28 mild COVID-19 patients, and 38 severe COVID-19 patients). (**B**) Pre-defined marker expression classification matrix used for BinaryClust. (**C**) *t*-Distributed stochastic neighbour embedding (*t*-SNE) plots, with 1000 cells per sample, were coloured by the main cell types generated by BinaryClust and faceted by different study groups. (**D**) The corresponding median marker expression heatmap of BinaryClust results for the COVID-19 dataset.

used Mann–Whitney–Wilcoxon test corrected by Holm method. We then selected nine markers (IL-6, PD-L1, VISTA, IDO, TIM-3, TMEM173, Granzyme B, PPARg, and Ki-67) as the state markers which can reflect the functional or proliferative status of the immune cell types. Notably, as shown in ***Figure 7C***, Granzyme B was observed to be significantly highly expressed in COVID-19 patients versus healthy control across all identified immune cell types (all p < 0.05), TMEM173 was substantially up-regulated in COVID-19 patients particularly in monocytes, neutrophils, and CD4 T cells; and PD-L1 expression remained low for all cells, indicating the immune system of COVID-19 patients was highly activated without exhaustion regardless of disease severity, which was not explored previously. Since monocytes and neutrophils were of particular interest in the original paper, and the panel also included specific markers for in-depth interrogation, we then extracted the two population and carried out Phenograph (*k* = 60) to investigate the subclusters. As shown in ***Figure 7D–E***, Phenograph returned 16 and 14 subclusters, respectively, for monocytes and neutrophils, with heatmap provided in ***Figure 7—figure supplement 1A, B***.

Reanalysis of the COVID-19 dataset demonstrated the concordance of ImmCellTyper pipeline with original reports with additional findings, as well as its versatile interactive data visualisation functionalities.

## Discussion

In this study, we present an analytical pipeline named ImmCellTyper for systematic exploration of CyTOF data. This pipeline addresses a comprehensive range of analytical needs encompassing data quality check, batch effects examination/correction, cell type identification, and downstream differential analysis accompanied by high-quality, publishable data visualisations. Furthermore, ImmCellTyper

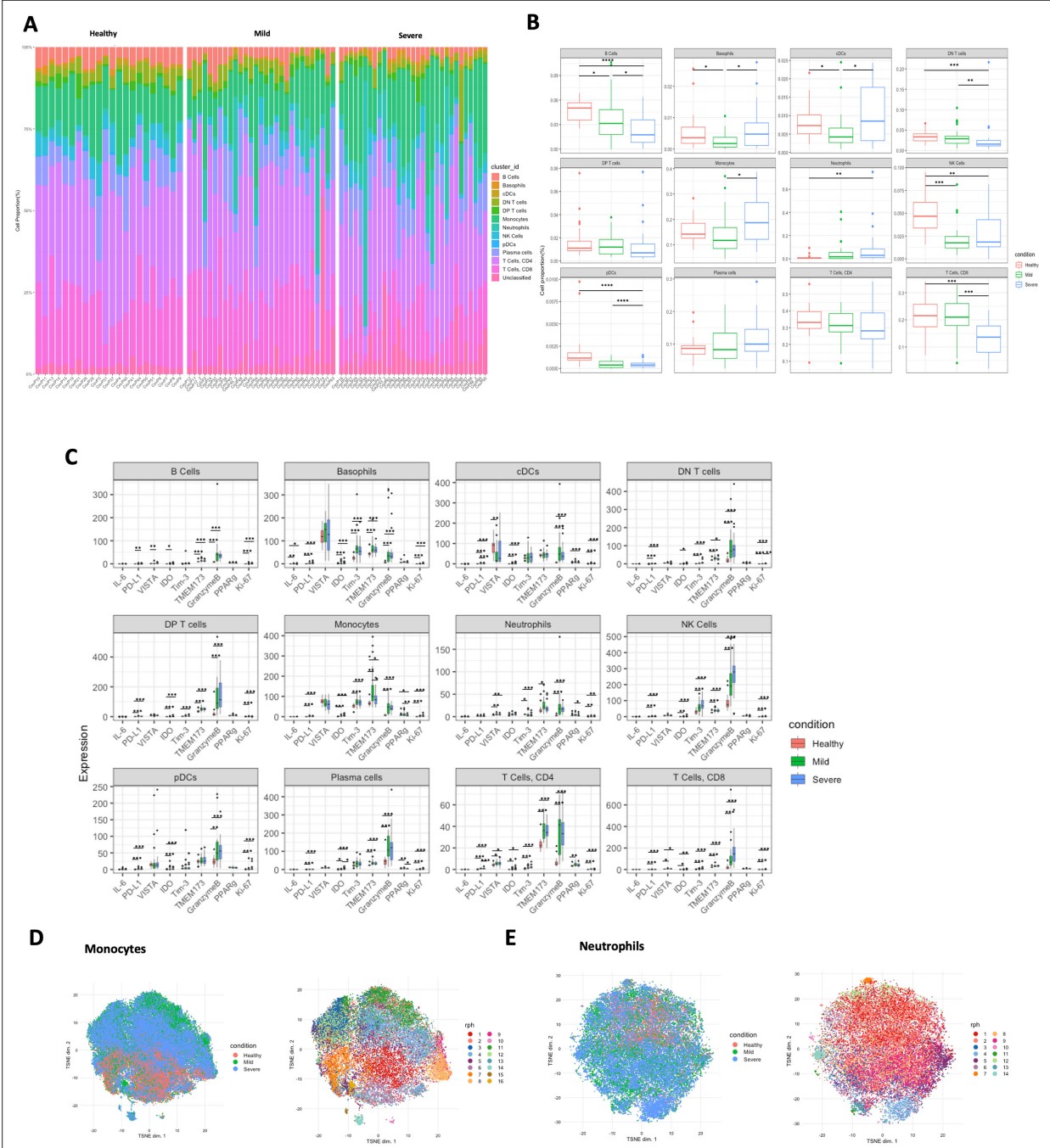

**Figure 7.** Quantification and statistical analysis comparing the study conditions in COVID-19 dataset (*n* = 82). (**A**) Stacked histogram of main cell type composition per individual generated by BinaryClust, and grouped by study conditions (healthy, mild, and severe). (**B**) Boxplots representing cell abundance frequencies among the study conditions, faceted by different main cell types. (**C**) State marker expression intensities with comparison of the study groups across the main cell types. (**D**) Clusters of monocytes and neutrophils were extracted from the whole cells for downstream interrogation. *t*-Distributed stochastic neighbour embedding (*t*-SNE) plots with random downsample of 1000 monocyte cells and (**E**) neutrophils per sample were coloured by study conditions and Phenograph clustering results (*k* = 60), respectively. Statistical significance was marked by asterisk. *p < 0.05, **p < 0.01, ***p < 0.001, ****p < 0.0001.

The online version of this article includes the following figure supplement(s) for figure 7:

**Figure supplement 1.** Expression heatmaps of monocytes and neotrophils in COVID-19 dataset.

includes an in-house developed, knowledge-based, semi-supervised classifier BinaryClust with high accuracy and speed, and also integrates the well-performing and state-of-the-art unsupervised algorithms (Phenograph and flowSOM). By adopting the strategy of first obtaining the main cell types and then select specific cell types for in-depth interrogation with higher clustering resolution, ImmCellTyper combines the advantages of both supervised and unsupervised clustering algorithms for discovery of both major populations and refined subpopulations. Designed for ease of use, this pipeline features a clear and user-friendly workflow, and is compatible to the widely used pipeline CATALYST, aiming to provide a one-stop solution for CyTOF users.

For validating the robustness of the automated cell characterisation function of ImmCellTyper: BinaryClust, we involved two independent datasets (MPN and Influenza datasets) and compared with existing clustering tools flowSOM and LDA. Previous benchmarking studies indicated that flowSOM was one of the best-performing unsupervised tools in precision, speed, and stability, with *F*-measure ranges from 0.58 to 0.90 in various datasets (*Liu et al., 2019*), and was widely used in high-impact research publications (*Liu et al., 2019*; *Liu et al., 2020*). However, the results of flowSOM vary upon the *k* values set by the users which causes some uncertainty and impairs reproducibility of the results. In contrast, semi-supervised clustering methods like ACDC, DeepCyTOF (*Li et al., 2017*), and LDA outperformed the unsupervised methods in the same benchmarking study. The *F*-measure obtained by ACDC and LDA ranged from 0.78 to 0.99, which was significantly higher than unsupervised approaches including Accense, PhenoGraph, Xshift, *k*-means, flowMeans (*Aghaeepour et al., 2011*), FlowSOM, and DEPECHE (*Theorell et al., 2019*) (*F*-measure: 0.28–0.93) (*Liu et al., 2020*). In this study, we chose LDA because it has proven to be superior to ACDC and DeepCyTOF in terms of precision and speed (*Liu et al., 2019*; *Abdelaal et al., 2019*). The excellent performance of LDA was demonstrated in MPN dataset of this study with *F*-measure reached 0.93 for around 4 million cells. BinaryClust was comparable in accuracy (*F*-measure = 0.94) but much faster in speed. Both semi-supervised approaches outperformed flowSOM as expected. Although semi-supervised methods seem to perform better, the mainstream for CyTOF data analysis still relies on unsupervised methods. The reason for this might be that most of the semi-supervised methods rely on manual gating as reference and requires either the manually gated expression matrix or user-defined binary matrix as training data. This process takes extra time and efforts especially for defining subpopulations, and may introduce end-user bias. Additionally, the common restriction for all the supervised methods is that they have limited capability to reveal novel subsets as those are not pre-defined in the training pool, which poses a critical challenge for the pursuit of novel discoveries. Considering all these strength and limitations, as a semi-supervised method, BinaryClust possesses the advantages of incorporating prior biological knowledge, being reproducible across runs and accurate on cell type identification. Meanwhile, the input requirement is a simple matrix of cell population definition based on marker positivity, avoiding the hassle to manually gate example FCS files. With a deep understanding that certain CyTOF markers are expressed on a continuum rather than binary manner, this automated supervised approach will only be used for classification of main cell lineages. The test datasets in this study contained data from human PBMC samples which represented a heterogeneous pool of immune cells, covering the major types of various immune populations, and demonstrated excellent performance of BinaryClust on classifying CD4 T cells, CD8 T cells, gamma delta T cells, NK cells, monocytes, dendritic cells, and B cells. We also recommend checking marker expression distribution before running the pipeline to ensure accurate results. After the above step, this pipeline supports cell population extraction for further dissection into subclusters using unsupervised approaches, which excel in detecting rare and refined subpopulations. Phenograph is a particularly effective tool for this purpose. By applying this strategy, our previous work with BinaryClust facilitated the evaluation of the impact of systematic anti-cancer agents on lymphocyte population in non-small cell lung cancer patients, as well as the characteristics of autologous T cell products after manufacturing (*O'Brien Gore et al., 2023*).

Another purpose behind the development of the ImmCellTyper framework is to cater to general analysis needs, spanning from pre-processing to downstream differential analysis. There are existing well-established pipelines like CATALYST (*Nowicka et al., 2017*), diffcyt (*Weber et al., 2019*), Cytofkit (*Chen et al., 2016*), ImmunoCluster (*Opzoomer et al., 2021*), etc., each of them has its own strength and limitations. One of the challenges existed in CyTOF data pre-processing is effectively integrating multiple batches, given the technical differences arising from experiments and instrumental acquisition,

which could result in the separation of clustering across batches and potentially confounding the signal of interest. In ImmCellTyper pipeline, we provide the functionality for batch effects examination and incorporated two well-performing batch correction algorithms CytoNorm (*Van Gassen et al., 2020*) and CytofRUV (*Trussart et al., 2020*) into our framework prior to downstream analysis, which ensures the quality of the data and, to the best of our knowledge, makes it the first integrated pipeline with this module. For the downstream analysis, tools like diffcyt and CATALYST group markers into phenotypic and state makers, enabling the detection of differentially abundant cell clusters and differential expression of functional markers within each cell population (*Weber et al., 2019*). This well-recognised strategy has been extensively applied in various studies. But one limitation is that it directly classifies cells into high-resolution clusters, and cannot automatically merge subclusters into one major population with similar phenotypes (*Weber et al., 2019*). Our pipeline addresses this problem via a precise and convenient solution: using the semi-supervised classifier (BinaryClust) of ImmCellTyper. The two pipelines (CATALYST and ImmCellTyper) are well compatible, allowing users to leverage their functions together and providing a broader range of analytical options. On the other hand, Cytofkit has the advantages of the graphical user interface for non-specialists, integration of a variety of clustering (DensVM [*Becher et al., 2014*], ClusterX, FlowSOM, Phenograph), dimension reduction (PCA, ISOMAP, *t*-SNE) methods, and inference of the relatedness among cell populations, but it does not support complex study design, group comparison, and statistical testing (*Chen et al., 2016*). Several methods have been developed to fit with the high-dimensional features of CyTOF data for differential analysis including diffcyt, CellCnn (*Arvaniti and Claassen, 2017*), cydar (*Lun et al., 2017*), citrus (*Bruggner et al., 2014*), cyEMD (*Arend et al., 2022*), etc. Nonetheless, none of them is applicable to compare more than two study groups at once which involves multiple testing. Therefore, to accommodate study designs involving multiple groups, ImmCellTyper does statistics via first applying Kruskal–Wallis test, followed by BH procedure for multiple testing correction and post hoc analysis via Dunn's test or pairwise Wilcoxon test. In cases where the comparison involves only two groups, Mann–Whitney test will be applied in terms of cell frequency, as the distribution of CyTOF data generally does not fit for normal distribution.

One limitation of this computational pipeline is that its semi-supervised cluster identification may not be well suited for the direct identification of specific subpopulations which are defined by continuum markers. This is due to the method's reliance on the presumption that makers are binary distributed. In addition, for users with general interest in charactering subclusters of each main cell lineage, it can be laborious to first perform BinaryClust then extract every population for unsupervised clustering. In such scenarios, BinaryClust can be used in parallel with other clustering methods or pipelines like CATALYST. This allows users to quickly obtain additional information about main cell types with the accuracy comparable to manual gating.

In summary, we introduce a novel open-source R-implemented strategy and a versatile toolbox for CyTOF data analysis. The future direction is to automate the data clean-up, compensation, and bead normalisation steps. Additionally, we also aim to implement the whole pipeline into Python, a more accessible programming language for bioinformatics novices and biologists who wish to perform high-dimensional data analysis independently.

## Materials and methods

**Key resources table**

| Reagent type (species) or resource | Designation | Source or reference | Identifiers | Additional information |
|---|---|---|---|---|
| Antibody | Maxpar Direct Immune Profiling Assay | Standard Biotools | Cat# 201325 | |
| Antibody | anti-CD95 (RRID:AB_314546, mouse, monoclonal) | Biolegend | Cat# 305607 | 1:200, 1.5 µl |
| Antibody | anti-TIGIT (mouse, monoclonal) | Standard Biotools | Cat# 201406 (Maxpar Direct T cell expansion panel 2) | 1:300, 1 µl |
| Antibody | anti-PD1 (mouse, monoclonal) | Standard Biotools | Cat# 201406 (Maxpar Direct T cell expansion panel 2) | 1:200, 1.5 µl |

*Continued on next page*

*Continued*

| Reagent type (species) or resource | Designation | Source or reference | Identifiers | Additional information |
|---|---|---|---|---|
| Antibody | anti-ICOS (hamster, monoclonal) | Standard Biotools | Cat# 201406 (Maxpar Direct T cell expansion panel 2) | 1:200, 1.5 µl |
| Antibody | anti-TIM3 (mouse, monoclonal) | Standard Biotools | Cat# 201406 (Maxpar Direct T cell expansion panel 2) | 1:300, 1 µl |
| Antibody | anti-OX40 (mouse monoclonal) | Standard Biotools | Cat# 201406 (Maxpar Direct T cell expansion panel 2) | 1:300, 1 µl |
| Antibody | anti-CXCR4 (mouse, monoclonal) | Standard Biotools | Cat# 201406 (Maxpar Direct T cell expansion panel 2) | 1:300, 1 µl |
| Genetic reagent; Recombinant DNA reagent | Pierce 16% Formaldehyde (methanol-free) | Thermo Fisher | Cat# 28906 | |
| Genetic reagent; Recombinant DNA reagent | Trypan blue solution (0.4%) | Gibco | Cat# 15250-061 | |
| Genetic reagent; Recombinant DNA reagent | Human TruStain FcX | Biolegend | Cat# 422302 | |
| Genetic reagent; Recombinant DNA reagent | EQ Four Element Calibration beads | Standard Biotools | Cat# 201078 | |

## BinaryClust

Most of the CyTOF markers exhibit log-normal or bi-modal distribution with zero inflation after arcsinh transformation. Thus, we employ binary classification using *k*-means to group cells into negative and positive populations (*k* = 2) for each marker indicated in the user-defined classification matrix. Here, *k*-means is an unsupervised clustering algorithm to cluster the data based on the Euclidean distance among points, which is calculated by the formula below:

$$d(p,q) = \sqrt{(p_1 - q_1)^2 + (p_2 - q_2)^2 + \cdots + (p_n - q_n)^2}.$$

Then assign each data point into a cluster centroid which is denoted by *ci*, and dist() is the Euclidean distance:

$$\arg \min_{ci \in C}(ci, x)^2$$

By aligning the *k*-means results with the user-designed classification matrix, cell populations can be subsequently classified and annotated.

## R package ImmCellTyper

This computational pipeline is implemented in the R package ImmCellTyper and publicly available on Github (https://github.com/JingAnyaSun/ImmCellTyper, *Sun et al., 2023*). Instructions for package installation and function usage can be found in the README file on the Github page. It is recommended to first examine the unwanted non-biological variation across batches and perform additional batch normalisation if necessary. Afterwards, import the data into the second part of the pipeline for downstream analysis. Users are required to prepare all the FCS files, sample metadata containing the details and grouping information of each sample, panel metadata with the information of the antibody panel with metal tags used in the experiments, and cell type classification matrix with phenotypic marker expression in a binary manner of each cell lineage. All files need to be in the right format to use the pipeline.

| Workflow steps | Description |
|---|---|
| Data clean-up and pre-processing | Bead normalisation is first performed on the raw FCS files to correct signal fluctuation of the CyTOF instrument, then FCS files are imported to cytobank for clean-up to exclude doublets and debris etc. |
| Batch effect evaluation and correction | Samples across different batches will be evaluated on both marker expression level and clustering level for batch effects; If needed, batch correction will be performed using function batchNorm, which provides two well-performing algorithms CytoNorm and CyTOFRUV. |
| Data transformation and SCE object construction | FCS files and relevant metadata for(samples and panel) will be integrated into one sce object, and FCS data will be transformed using co-factor 5. |
| Binary classification | BinaryClust will be subsequently performed based on the classification matrix designed by the user. |
| Differential analysis and population extraction | Differential cell abundance analysis and statistical comparison will be conducted. Then the user can quickly gate certain population of interest for down-stream in-depth analysis. |
| In-depth interrogation of population of interest | Unsupervised clustering such as flowSOM and Phenograph can be implemented for the extracted population for further clustering. |
| Differential analysis and statistical comparison | In this pipeline we support multiple study group statistical analysis(n>2) with multiple testing correction and post hoc analysis. |

**Figure 8.** Overall schematic outline of the ImmCellTyper workflow with description for each step.

## ImmCellTyper workflow overview

ImmCellTyper pipeline is composed of seven steps corresponding to two separate sub-pipelines, as described in *Figure 8* and Github vignettes (https://github.com/JingAnyaSun/ImmCellTyper/tree/main/vignettes; *Sun et al., 2023*):

1. Sub-pipeline1 (corresponds to *Figure 8*, workflow step 2): Batch effects evaluation and correction. Batch effects occur when samples were collected and measured at different sites or time points, especially for large-scale studies. It is crucial to remove the unwanted variation which might interfere the true biological differences. After bead normalisation and data clean-up (*Figure 8*, workflow step 1), which can be performed using third-party platforms, such as CyTOF v7.0 system control software and Cytobank, users can systematically examine batch effects on two levels including marker behaviours and clustering results based on the method introduced by *Trussart et al., 2020*. If needed, CytoNorm and CytofRUV, which are well-performing correction algorithms, can be used to align the existing batch effects.

2. Sub-pipeline2 (corresponds to *Figure 8*, workflow steps 3–7): Semi-supervised classification, differential analysis, and in-depth investigation. When data are cleaned and normalised, they can be imported into the second part of the pipeline, constructed into a SingleCellExperiment object, and undergo semi-supervised classification to identify the major cell types and test the differential frequencies or state marker expression among study groups. After that, if the users have certain interests of specific cell types and pre-design the panel for that, or in another circumstance, the initial statistics draw the user's attention into a certain cell type, institutively, further clustering using unsupervised tools like flowSOM or Phenograph should be conducted with an increased cluster resolution and deeper investigation for cell subsets, after extracting the cell population of interest. ImmCellTyper pipeline has the same data storage and infrastructure as CATALYST, therefore all the functions in CATALYST can be seamlessly used in ImmCellTyper, tailored to user's analytical needs. We do not elaborate on the basic functions of CATALYST, which can be found in the tutorial vignettes of the package (https://github.com/HelenaLC/CATALYST; *CATALYST-project, 2024*).

## Batch correction algorithms

The batch effect correction algorithms embedded in the function of 'batchNorm' comprises CytoNorm, as described by *Van Gassen et al., 2020* and CytofRUV by *Trussart et al., 2020*. Both algorithms rely on anchors (reference samples/technical replicates) across batches to perform normalisation. CytoNorm uses flowSOM clustering to first identify clusters prior to a population-specific

transformation on the reference samples by computing the quantile values and aligning them with splines, whereas CytofRUV applies remove unwanted variation III (RUV-III) to CyTOF data by estimating and eliminating the non-biological variation of the pseudo-replicates.

## Agreement evaluation

To assess the agreement among manual gating, BinaryClust classification and flowSOM clustering, we used correlation, interaction plot, and Bland–Altman analysis. Correlation evaluates the relationship between two variables which does not mean concordance, but if two methods agree, surely, they should be highly correlated. We compared the results of cell population frequency generated by experts and BinaryClust using Pearson correlation, calculated the correlation coefficient and p value, with the line of equality indicating perfect agreement (red solid line, $R^2 = 1$) in the plots to help gauge the degree of agreement between the two methods; Bland–Altman plot refers to a dot plot of the difference between two variables (y-axis) against the mean of them (x-axis), as described by J. Martian Bland and Douglas G. Altman in 1986, which represents a graphical magnitude of bias (average of difference) with 95% confidence interval. The math formula for the limit of agreement is as below:

$$\text{Limits of agreement} = \text{ mean difference observed} \pm 1.96 \times \text{standard deviation}$$

We also used Interaction plot to display the interaction effects of the three methods including manual gating, BinaryClust classification, and flowSOM clustering on the measurement of cell frequencies to evaluate the agreement.

## *F*-measure

*F*-measure is a method to assess the accuracy of the clustering method compared to gold standard, which is manual gating results in this study. It stands for the harmonic mean of the precision and recall values, which can be calculated using below formula:

$$F-\text{measure} = 2x \frac{\text{Precision} \times \text{Recall}}{\text{Precision} + \text{Recall}}$$

Here, precision represents the positive predictive value: the proportion of true positive instances divided by the instances classified as positive by the clustering algorithm; recall evaluates the sensitivity, which is the number of true positive events correctly identified by the algorithm among all events that belong to the cluster. *F*-measure ranges from 0 to 1, where 1 suggests perfect performance and 0 indicates poor precision and recall.

## Adjusted Rand Index

Adjusted Rand Index is a widely used method to measure the similarity between two clustering results. We used it to assess the agreement between test clustering algorithm and the gold standard labels which is derived from manual gating. ARI is defined as the following formula based on a contingency matrix (where $n_{ij}$, $a_i$, $b_j$ are values from the contingency table):

$$\text{ARI} = \frac{\Sigma_{ij} \binom{n_{ij}}{2} - \left[ \Sigma_i \binom{a_i}{2} \Sigma_i \binom{b_i}{2} \middle/ \binom{n}{2} \right]}{\frac{1}{2} \left[ \Sigma_i \binom{a_i}{2} + \Sigma_j \binom{b_i}{2} \right] - \left[ \Sigma_i \binom{a_i}{2} + \Sigma_j \binom{b_i}{2} \right] \middle/ \binom{n}{2}}$$

## Differential analysis

In order to perform the differential analysis of cell abundances and state marker expression among study groups, we consider the number of study groups into two conditions in function 'StatTest': (1) $n$ = 2, Mann–Whitney–Wilcoxon analysis will be applied as cell frequency/marker expression does not fit normal distribution; (2) $n$ = 3, Kruskal–Wallis test will be first performed followed by multiple testing correction using BH procedure and post hoc analysis (Dunn's test or pairwise Wilcoxon test).

## Sample collection and preparation for CyTOF

PBMC samples from MPN patients were requested and obtained from biobank at Guy's Hospital, under a protocol approved by the KCL Biobank Access Committee (REC18/EE/0025). Healthy volunteers were recruited at Guy's hospital with informed consent and ethical approval by King's College Research Ethics Committee (HR-17/18-5960 MOD-20/21-5960) in accordance with the Declaration of Helsinki. All identifiable information of study participants were securely stored in a trusted research environment managed by members of the team.

Venous blood samples from healthy volunteers were collected in BD vacutainer Ethylenedi-aminetetraacetic Acid (EDTA) tubes, and PBMCs were isolated and purified using Ficoll-Hypaque density gradient centrifugation. In brief, blood was carefully layered onto Ficoll and spined at 460 × *g* for 20 min at room temperature (RT) without brake; upon observing a clear separation of blood components, PBMCs were then carefully isolated and washed three times with RPMI 1640 medium to remove other contaminants. Subsequently, the purified PBMCs were cryopreserved in liquid nitrogen at a density of $1 \times 10^7$ cells per vial.

## CyTOF antibody staining

Cryopreserved cells were quickly thawed at 37°C in water bath, suspended in pre-warmed RPMI 1640 medium, washed three times, and underwent Fc-blocking using human TruStain FcX (Biolegend) for 10 min at RT. Given that the antibody panel employs indirect detection of CD95 using anti-CD95-APC and anti-APC-106Cd, cells were initially stained with anti-CD95-APC for 30 min in a dark place, followed by two washes with 2-ml cell staining buffer (CSB). After the removal of supernatant, cells were resuspended in 300 µl CSB and transferred into the dip tube containing lyophilised antibody mix of Maxpar direct immune profiling assay (Standard BioTools). Additional antibodies used to study T cell activation, migration, and exhaustion status (anti-TIM3, anti-PD-1, anti-ICOS, anti-TIGIT, and anti-OX40) were also added. Cells were incubated with the antibody mix for 30 min at RT in compliance with the manufacturer's instructions, washed twice and fixed using a freshly prepared 1.6% paraformaldehyde (PFA, Thermo Fisher Scientific) solution for 10 min at RT. In the end, cells were washed with CSB twice to remove residual PFA and stained with 125 nM Cell-ID intercalator-Iridium in 1 ml Maxpar Fix and Perm Buffer (Standard BioTools) overnight at 4°C. The stained cells were frozen down using freezing media (fetal bovine serum containing 10% Dimethyl sulfoxide (DMSO)) and stored at −80°C freezer before data acquisition.

On the day for CyTOF acquisition, cryopreserved samples were thawed, washed twice with 1 ml CSB, followed by additional two washes with 1 ml cell acquisition solution (Standard BioTools), and centrifuged at 800 × *g* for 5 min. Cells were then filtered through a 40-µm cell strainer to avoid blockage and cell count was determined by Countess automated cell counter (Invitrogen). EQ four-element calibration beads (Standard BioTools) were added to a final concentration of $0.5 \times 10^6$ cells/ml to adjust signal fluctuation of the instrument. CyTOF acquisition was performed on Helios mass cytometer system. For each batch of cell staining and run, technical replicates from the same healthy donor were included to evaluate and correct batch effects.

The full antibody panel including the metal tag, clone, and supplier is listed in *Appendix 1—table 1*.

## CyTOF data pre-processing

FCS files were first processed for bead normalisation using CyTOF v7.0 system control software (Standard BioTools) to correct signal drift during acquisition. Subsequently, the files were imported into Cytobank (Cytobank Inc) for data cleaning, with a detailed procedure illustrated in *Figure 1—figure supplement 2*. The aim is to remove non-events including debris, doublets, normalisation (EQ) beads, and other undesired events like dead cells. The CD45+ cell population was pre-gated using Cytobank for downstream analysis.

## Manual gating

Manual gating for major cell populations was performed using Cytobank platform, and gatingML files were exported and converted into gatingSet object to extract labels of each cell via the R packages flowWorkspace, openCyto, and CytoML. These data were used as reference for benchmarking computer-aided algorithms.

## Acknowledgements

This work is supported in part by: Blood Cancer UK [Ref. 22009] and CRUK City of London Centre (CoL) Award [CTRQQR-2021/100004] to MM and SK; JS was supported by the K-CSC scholarship. We thank BRC flowCore at Guy's Hospital for technical support, Biobank at King's College London for patient samples; We are thankful for Cynthia Bishop, Katrina Todd, Rosa Andres Ejarque, Elena Torre, Robert Page, Rita Antunes Dos Reis, and Mishto lab members at King's College London, Daniel Dancer, Thomas Adejumo at Standard Biotools for technical assistance and sample management, as well as all study participants for their kind support for this research. Special thanks to Eliezer Van Allen and Jihye Park at Dana Farber Cancer Institute to review and comment this work.

## Additional information

### Funding

| Funder | Grant reference number | Author |
|---|---|---|
| Blood Cancer UK | Ref. 22009 | Michele Mishto<br>Shahram Kordasti |
| Cancer Research UK | CTRQQR-2021/100004 | Michele Mishto<br>Shahram Kordasti |
| China Scholarship Council | | Jing Sun |

The funders had no role in study design, data collection, and interpretation, or the decision to submit the work for publication.

### Author contributions

Jing Sun, Conceptualization, Data curation, Software, Formal analysis, Investigation, Visualization, Methodology, Writing - original draft, Writing - review and editing; Desmond Choy, Conceptualization, Software, Supervision, Investigation, Methodology, Writing - review and editing; Nicolas Sompairac, Validation, Methodology, Writing - review and editing; Shirin Jamshidi, Investigation, Methodology, Writing - review and editing; Michele Mishto, Supervision, Funding acquisition, Project administration, Writing - review and editing; Shahram Kordasti, Conceptualization, Resources, Supervision, Funding acquisition, Investigation, Project administration, Writing - review and editing

### Author ORCIDs

Jing Sun ![ORCID] https://orcid.org/0009-0001-7102-7556
Nicolas Sompairac ![ORCID] https://orcid.org/0000-0001-8759-2077
Shirin Jamshidi ![ORCID] https://orcid.org/0000-0001-8290-6698
Shahram Kordasti ![ORCID] https://orcid.org/0000-0002-0347-4207

### Ethics

PBMC samples from MPN patients were requested and obtained from biobank at Guy's Hospital, under a protocol approved by the KCL Biobank Access Committee (REC18/EE/0025). Healthy volunteers were recruited at Guy's hospital with informed consent and ethical approval by King's College Research Ethics Committee (HR-17/18-5960 MOD-20/21-5960) in accordance with the Declaration of Helsinki. All identifiable information of study participants were securely stored in a trusted research environment managed by members of the team.

Reviewer #3 (Public rbioreview): https://doi.org/10.7554/eLife.95494.3.sa1
Author response https://doi.org/10.7554/eLife.95494.3.sa2

## Additional files

### Supplementary files
• MDAR checklist

## Data availability

In this study, we tested ImmCellTyper pipeline on the MPN cohort with 7 MPN patients and 2 healthy volunteers, influenza cohort with 11 patients, and the COVID-19 cohort with 59 COVID-19 patients and 23 healthy volunteers. The FCS files (after clean-up and gating of CD45 population) and metadata of the MPN cohort were deposited in Zenodo (https://doi.org/10.5281/zenodo.10076940); the influenza cohort was published by our lab (*Alimam et al., 2021*), and the data were stored in Zenodo (https://doi.org/10.5281/zenodo.7982165); the COVID-19 dataset was previously published by Chevrier et al. (*Chevrier et al., 2021*) and the FCS files can be retrieved from Mendeley Data (https://data.mendeley.com/datasets/vyy8ttw7n9/1).

The following dataset was generated:

| Author(s) | Year | Dataset title | Dataset URL | Database and Identifier |
| --- | --- | --- | --- | --- |
| Sun J | 2023 | CyTOF data for MPN dataset | https://zenodo.org/records/10076940 | Zenodo, 10.5281/zenodo.10076940 |

The following previously published datasets were used:

| Author(s) | Year | Dataset title | Dataset URL | Database and Identifier |
| --- | --- | --- | --- | --- |
| Chevrier S, Zurbuchen Y, Cervia C | 2020 | A distinct innate immune signature marks progression from mild to severe COVID-19 | https://doi.org/10.17632/vyy8ttw7n9.1 | Mendeley Data, 10.17632/vyy8ttw7n9.1 |
| Alimam S | 2021 | CyTOF test data for BinaryClust2 | https://doi.org/10.5281/zenodo.7982165 | Zenodo, 10.5281/zenodo.7982165 |

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

# Appendix 1

**Appendix 1—table 1.** CyTOF antibody panel for MPN cohort.

| Antibody | Clone | Metal tag | Manufacturer |
|---|---|---|---|
| Anti-CD45 | HI30 | 89Y | Standard BioTools |
| Live/dead Indicator | N/A | 103Rh | Standard BioTools |
| Anti-CD95 | DX2 | APC | Biolegend |
| Anti-APC | APC003 | 106Cd | Standard BioTools |
| Anti-CD196 | G034E3 | 141Pr | Standard BioTools |
| Anti-OX40 | ACT35 | 142Nd | Standard BioTools |
| Anti-CD123 | 6H6 | 143Nd | Standard BioTools |
| Anti-CD19 | HIB19 | 144Nd | Standard BioTools |
| Anti-CD4 | RPA-T4 | 145Nd | Standard BioTools |
| Anti-CD8a | RPA-T8 | 146Nd | Standard BioTools |
| Anti-CD11c | Bu15 | 147Sm | Standard BioTools |
| Anti-CD16 | 3G8 | 148Nd | Standard BioTools |
| Anti-CD45RO | UCHL1 | 149Sm | Standard BioTools |
| Anti-CD45RA | HI100 | 150Nd | Standard BioTools |
| Anti-CD161 | HP-3G10 | 151Eu | Standard BioTools |
| Anti-CD194 | L291H4 | 152Sm | Standard BioTools |
| Anti-CD25 | BC96 | 153Eu | Standard BioTools |
| Anti-CD27 | O323 | 154Sm | Standard BioTools |
| Anti-CD57 | HCD57 | 155Gd | Standard BioTools |
| Anti-CD183 | G025H7 | 156Gd | Standard BioTools |
| Anti-CD185 | J252D4 | 158Gd | Standard BioTools |
| Anti-TIM-3 | F38-2E2 | 159Tb | Standard BioTools |
| Anti-CD28 | CD28.2 | 160Gd | Standard BioTools |
| Anti-CD38 | HB-7 | 161Dy | Standard BioTools |
| Anti-CD56 | NCAM16.2 | 163Dy | Standard BioTools |
| Anti-TCRgd | B1 | 164Dy | Standard BioTools |
| Anti-PD-1 | EH12.2H7 | 165Ho | Standard BioTools |
| Anti-CD294 | BM16 | 166Er | Standard BioTools |
| Anti-CD197 | G043H7 | 167Er | Standard BioTools |
| Anti-CD14 | 63D3 | 168Er | Standard BioTools |
| Anti-ICOS | C398.4A | 169Tm | Standard BioTools |
| Anti-CD3 | UCHT1 | 170Er | Standard BioTools |
| Anti-CD20 | 2H7 | 171Yb | Standard BioTools |
| Anti-CD66b | G10F5 | 172Yb | Standard BioTools |
| Anti-HLA-DR | LN3 | 173Yb | Standard BioTools |
| Anti-IgD | IA6-2 | 174Yb | Standard BioTools |
| Anti-CXCR4 | 12G5 | 175Lu | Standard BioTools |

*Appendix 1—table 1 Continued on next page*

*Appendix 1—table 1 Continued*

| Antibody | Clone | Metal tag | Manufacturer |
| --- | --- | --- | --- |
| Anti-CD127 | A019D5 | 176Yb | Standard BioTools |
| Anti-TIGIT | MBSA43 | 209Bi | Standard BioTools |

