## [Editor Report · eLife assessment]

ImmCellTyper presents a **useful** toolkit for CyTOF data analysis, integrating BinaryClust for semi-supervised clustering and cell type annotation. The evidence supporting the findings is **convincing**, with appropriate and validated methodology. This tool will be helpful to researchers in immunology and cytometry, offering a robust solution for cell type identification and differential analysis.

---

## [Referee Report · Reviewer #3 (Public rbioreview)]

Summary:

ImmCellTyper is a new toolkit for Cytometry by time-of-flight data analysis. It includes BinaryClust, a semi-supervised clustering tool (which takes into account the prior biological knowledge), designed for automated classification and annotation of specific cell types and subpopulations. ImmCellTyper also integrates a variety of tools to perform data quality analysis, batch effect correction, dimension reduction, unsupervised clustering, and differential analysis.

Strengths:

The proposed algorithm takes into account the prior knowledge.

The results on different benchmark indicates competitive or better performance (in terms of accuracy and speed) depending on the method.

---

## [Author Response]

The following is the authors’ response to the original reviews.

**Public Reviews:**

**Reviewer #1 (Public Review):**
Summary:This manuscript presented a useful toolkit designed for CyTOF data analysis, which integrates 5 key steps as an analytical framework. A semi-supervised clustering tool was developed, and its performance was tested in multiple independent datasets. The tool was compared to human experts as well as supervised and unsupervised methods.Strengths:The study employed multiple independent datasets to test the pipeline. A new semi-supervised clustering method was developed.Weaknesses:The examination of the whole pipeline is incomplete. Lack of descriptions or justifications for some analyses.

We thank the reviewer’s overall summary and comments of this manuscript. In the last part of the results, we showcased the functionalities of ImmCellTyper in covid dataset, including quality check, BinaryClust clustering, cell abundance quantification, state marker expression comparison within each identified cell types, cell population extraction, subpopulation discovery using unsupervised methods, and data visualization etc. We added more descriptions in the text based on the reviewer’s suggestions.

**Reviewer #2 (Public Review):**
Summary:The authors have developed marker selection and k-means (k=2) based binary clustering algorithm for the first-level supervised clustering of the CyTOF dataset. They built a seamless pipeline that offers the multiple functionalities required for CyTOF data analysis.Strengths:The strength of the study is the potential use of the pipeline for the CyTOF community as a wrapper for multiple functions required for the analysis. The concept of the first line of binary clustering with known markers can be practically powerful.Weaknesses:The weakness of the study is that there's little conceptual novelty in the algorithms suggested from the study and the benchmarking is done in limited conditions.

We thank the reviewer’s overall summary and comments of this manuscript. While the concept of binary clustering by k-means is not novel, BinaryClust only uses it for individual markers to identify positive and negative cells, then combine it with the pre-defined matrix for cell type identification. This has not been introduced elsewhere. Furthermore, ImmCellTyper streamlines the entire analysis process and enhances data exploration on multiple levels. For instance, users can evaluate functional marker expression level/cellular abundance across both main cell types and subpopulations; Also, this computational framework leverages the advantages of both semi-supervised and unsupervised clustering methods to facilitate subpopulation discovery. We believe these contributions warrant consideration as advancements in the field.

As for the benchmarking, we limited the depth only to main cell types rather than subpopulations. The reason is because we only apply BinaryClust to identify main cell types; For the cell subsets discovery, unsupervised methods integrated in this pipeline has already been published and widely used by the research community. Therefore, it does not seem to be necessary for additional benchmarking.

**Reviewer #3 (Public Review):**
Summary:ImmCellTyper is a new toolkit for Cytometry by time-of-flight data analysis. It includes BinaryClust, a semi-supervised clustering tool (which takes into account prior biological knowledge), designed for automated classification and annotation of specific cell types and subpopulations. ImmCellTyper also integrates a variety of tools to perform data quality analysis, batch effect correction, dimension reduction, unsupervised clustering, and differential analysis.Strengths:The proposed algorithm takes into account the prior knowledge.The results on different benchmarks indicate competitive or better performance (in terms of accuracy and speed) depending on the method.Weaknesses:The proposed algorithm considers only CyTOF markers with binary distribution.

We thank the reviewer’s overall summary and comments of this manuscript. Binary classification can be considered as an imitation of human gating strategy, as it is applied to each marker. For example, when characterizing the CD8 T cells, we aim for CD19-CD14-CD3+CD4- population, which is binary in nature (either positive and negative) and follows the same logic as the method (BinaryClust) we developed. Results indicated that it works very well for well-defined main cell lineages, particularly when the expression of the defining marker is not continuous. However, the limitation is for subpopulation identification, because a handful makers behave in a continuum manner, so we suggest unsupervised method after BinaryClust, which also brings another advantage of identifying unknown subsets beyond our current knowledge, and none of the semi-supervised tools can achieve that. To address the reviewer’s concern, we considered the limitation of binary distribution, but it does not profoundly affect the application of the pipeline.

**Recommendations for the authors:**

**Reviewer #1 (Recommendations For The Authors):**

Many thanks for the reviewers’ comments and suggestions, please see below the point-to-point response:

(1) The style of in-text reference citation is not consistent. Many do not have published years.

The style of the reference citation has been revised and improved.

(2) The font size in the table of Figure 1 is too small, so is Figure 2.

The font size has been increased.

(3) Is flowSOM used as part of BinaryClust? How should the variable running speed of BinaryClust be interpreted, given that it is occasionally slower and sometimes faster than flowSOM in the datasets?

To answer reviewer’s question, flowSOM is not a part of BinaryClust. They are separate clustering methods that have been incorporated into the ImmCellTyper pipeline. As described in Figure 1, BinaryClust, a semi-supervised method, is used to classify the main cell lineages; while flowSOM, an unsupervised method, is recommended here for further subpopulation discovery. So, they operate independently of each other. To avoid confusions, we slightly modified Figure 1 for clarification.

Regarding the variability in running speed in Figure 4. The performance of algorithms can indeed be influenced by the characteristics of the datasets, such as size and complexity. The differences observed between the covid dataset and the MPN dataset, such as marker panel, experimental protocol, and data acquisition process etc., could account for this variation. Our explanation is that flowSOM suits better the data structure of covid dataset, which might be the reason why it is slightly faster to analyse compared to the MPN dataset. Moreover, for the covid dataset, the runtime for both BinaryClust and flowSOM is less than 100s, and the difference is not notable.

(4) In the Method section ImmCellTyper workflow overview, it is difficult to link the description of the pipeline to Figure 8. There are two sub-pipelines in the text and seven steps in the figure. What are their relations? Some steps are not introduced in the text, such as Data transformation and SCE object construction. What is co-factor 5?

Figure 8 provides an overview of the entire workflow for CyTOF data analysis, starting from the raw fcs file data and proceeding until downstream analysis (seven steps). But the actual implementation of the pipeline was divided into two separate sections, as outlined in the vignettes of the ImmCellTyper GitHub page (https://github.com/JingAnyaSun/ImmCellTyper/tree/main/vignettes).

Users will initially run ‘Intro_to_batch_exam_correct’ to perform data quality check and identify potential batch effects, followed by ‘Intro_to_data_analysis’ for data exploration. We agree with the reviewer that the method for this section is a bit confusing, so we’ve added more description for clarification.

In processing mass cytometry data, arcsine transformation is commonly applied to handle zero values, skewed distributions, and to improve visualization as well as clustering performance. The co-factor here is used as a parameter to scale down the data to control the width of the linear region before arcsine transformation. We usually get the best results by using co-factor 5 for CyTOF data.

(5) For differential analysis, could the pipeline analyze paired/repeated samples?

For the statistical step, ImmCellTyper supports both two-study group comparison using Mann-Whitney Wilcoxon test, and multiple study group comparison (n>2) using Kruskal Wallis test followed by post hoc analysis (pairwise Wilcoxon test or Dunn’s test) with multiple testing correction using Benjamini-Hochberg Procedure.

Certainly, this pipeline allows flexibilities, users can also extract the raw data of cell frequencies and apply suitable statistical methods for testing.

(6) In Figure 2A, the range of the two axes is different for Dendritic cells, which could be misleading. Why the agreement is bad for dendritic cells?

The range for the axes is automatically adapted to the data structure, which explains why they may not necessarily be equal. The co-efficient factor for the correlation of DCs is 0.958, compared to other cell types (> 0.99), it is relatively worse but does not indicate poor agreement.

Moreover, the abundance of DCs is much less than other cell types, comprising approximately 2-5% of whole cells. As a result, even small differences in abundance may appear to as significant variations. For example, a difference of 1% in DC abundance represents a 2-fold change, which can be perceived as substantial.

Overall, while the agreement for DCs may appear comparatively lower, it is not necessarily indicative of poor performance, considering both the coefficient factor and the relative abundance of DCs compared to other cell types.

(7) In the Results section BinaryClust achieves high accuracy, what method was used to get the p-value, such as lines 212, 213, etc.?

The accuracy of BinaryClust was tested using F-measure and ARI against ground truth (manual gating), the detailed description/calculation can be found in methods. For line 212 and 213, the p-value was calculated using ANOVA for the interaction plot shown in Figure 3. We’ve now added the statistical information into the figure legend.

(8) The performance comparison between BinaryClust and LDA is close. The current comparison design looks unfair. Given LDA only trained using half data, LDA may outperform BinaryClust.

It is true that LDA was trained using half data, which is because this method requires manual gating results as training dataset to build a model, then apply the model to the rest of the files to label cell types. Here we used 50% of the whole dataset as training set. We are of course very happy to implement any additional suggestions for a better partition ratio.

(9) There are 5 key steps in the proposed workflow. However, not every step was presented in the Results.

Thanks for the comments. The results primarily focused on demonstrating the precision and performance of BinaryClust in comparison with ground truth and existing tools. Additionally, a case study showcasing the application/functions of the entire pipeline in a dataset was also presented. Due to limitation in space, the implementation details of the pipeline were described in the method section and github documentations, which users/readers can easily access.

**Reviewer #2 (Recommendations For The Authors):**
The tools suggested by the authors could be potentially useful to the community. However, it's difficult to understand the conceptual novelty of the algorithms suggested here. The concept of binary clustering has been described before (https://doi.org/10.1186/s12859-022-05085-z, https://doi.org/10.1152/ajplung.00104.2022), and it mainly utilizes k-means clustering set to generate binary clusters based on selected markers. Other algorithms associated with the package are taken from other studies.

We acknowledge the reviewer’s comment regarding the novelty of our method. While the concept of binary clustering by k-means has been previously described to transcriptome data, our approach applies it to CyTOF data analysis, which has not been introduced elsewhere. Furthermore, ImmCellTyper streamlines the entire analysis process and enhances data exploration on multiple levels. For instance, users can evaluate functional marker expression level/cellular abundance across both main cell types and subpopulations; Also, as stated in the manuscript, this computational framework leverages the advantages of both semi-supervised and unsupervised clustering methods to facilitate subpopulation discovery. We believe these contributions warrant consideration as advancements in the field.

In addition, the benchmarking of clustering performance, especially to reproduce manual gating and comparison to tools such as flowSOM is not comprehensive enough. The result for the benchmarking test could significantly vary depending on how the authors set the ground truth (resolution of cell type annotations). The authors should compare the tool's performance by changing the depth of cell type annotations. Especially, the low abundance cell types such as gdT cells or DCs were not effectively captured by the suggested methods.

Thanks for the comment. We appreciate the reviewer’s concern. However, as illustrated in figure 1, our approach uses BinaryClust, a semi-supervised method, to identify main cell types rather than directly targeting subpopulations. The reason is because semi-supervised method relies on users’ prior definition thus is limited to discover novel subsets. In the ImmCellTyper framework, unsupervised method was subsequently applied for subset exploration following the BinaryClust step.

Regarding benchmarking, we focused on testing the precision of BinaryClust for main cell type characterization, because it is what the method is used for in the pipeline, and we believe this is sufficient. As for the cell subsets discovery, the unsupervised methods we integrated has already been published and widely used by the research community. Therefore, it does not seem to be necessary for additional benchmarking.

Moreover, as shown in Figure 3 and Table 1, our results indicated that the F-measure for DCs and gdT cells in BinaryClust is 0.80 and 0.92 respectively, which were very close to ground truth and outperformed flowSOM, demonstrating its effectiveness.

We hope these clarifications address the reviewer’s concern.

Minor comments:(1) In Figure 4, it's perplexing to note that BinaryClust shows the slowest runtime for the COVID dataset, compared to the MPN dataset, which features a similar number of cells. What causes this variation? Is it dependent on the number of markers utilized for the clustering? This should be clarified/tested.

Thanks for the comment, but we are not sure that we fully understand the question. As shown in figure 4 that BinaryClust has slightly higher runtime in MPN dataset than covid dataset, which is reasonable because and the cell number in MPN dataset is around 1.6 million more than covid dataset.

(2) Some typos are noted:- DeepCyTOF and LDA use a maker expression matrix extracted → "marker"?*

Corrected.

- Datasets(Chevrier et al.)which → spacing*

Corrected.

- This is due to the method's reliance → spacing*

Corrected.

**Reviewer #3 (Recommendations For The Authors):**
Is it possible to accommodate more than two levels within the clustering process, i.e., can the proposed semi-supervised clustering tool be extended to multi-levels instead of binary?

Thanks for the comments. Binary classification can be considered as an imitation of human gating strategy, as it is applied to each marker. For example, when characterizing the CD8 T cells, we aim for CD19-CD14-CD3+CD4- population, which is binary in nature (either positive and negative) and follows the same logic as the method (BinaryClust) we developed. Results indicated that it works very well for well-defined main cell lineages. However, the limitation is for subpopulation identification, because a handful of makers behave in a continuum manner, so we would suggest unsupervised method after BinaryClust, which also brings another advantage of identifying unknown subsets beyond our current knowledge, and none of the semi-supervised tools can achieve that. To answer the reviewer’s question, it is possible to set the number to 3,4,5 rather than just 2, but considering the design and rationale of the entire framework (as describe in the manuscript and above), it doesn’t seem to be necessary.

Could you please comment on why on the COVID dataset, BinaryClust was slower as compared to flowSOM?

Thanks for the question. The performance of algorithms can indeed be affected by the characteristics of the datasets, such as their size and complexity. The covid and MPN datasets differ in various aspects including marker panel, experimental protocol, and data acquisition process, among others, which wound account for the observed variation in speed. So, our explanation is flowSOM suits better for the structure of covid dataset than MPN dataset. Additionally, for covid dataset, both BinaryClust and flowSOM have runtimes of less than 100s, and the difference between the two isn’t particularly dramatic.

Minor errors:Line#215 "(ref) " reference is missing

Added.

Figure 3, increase the font of the text in order to improve readability.

Increased.

Line#229 didn't  did not.

Corrected

Line#293 repetition of the reference.

The repetition is due to the format of the citation, which has been revised.